# The Influence of Temperature on Metabolisms of Phosphorus Accumulating Organisms in Biological Wastewater Treatment Plants in the Presence of Cu(II) Toxicity

**Chen-Chiang Chou, Chih-Chi Yang, Meng-Shan Lu, Li-Yuan Hu, Ku-Fan Chen and Yung-Pin Tsai \***

Department of Civil Engineering, National Chi Nan University, 1, University Rd., Puli, Nantou 54561, Taiwan; ccc6166@yahoo.com.tw (C.-C.C.); s96322902@mail1.ncnu.edu.tw (C.-C.Y.); mslu@ncnu.edu.tw (M.-S.L.); whattodo38@gmail.com (L.-Y.H.); kfchen@ncnu.edu.tw (K.-F.C.)

**\*** Correspondence: yptsai@ncnu.edu.tw; Tel.: +886-49-2918230

**Abstract:** The purpose of this study was to study how temperature variation affects the tolerance of phosphorus accumulating organisms (PAOs) in a toxic environment. To exclude the interference of glycogen accumulating organisms (GAOs), shock loading experiments were conducted to study the effect of Cu(II) toxicity on the metabolisms of PAOs in 10, 20, and 30 °C conditions. The experimental data showed that the temperature effects on aerobic phosphorus uptake, PHA degradation, and glycogen synthesis were remarkable in the presence of Cu(II). Nevertheless, insignificant effects on anaerobic phosphorus release and PHA synthesis were found. The largest inhibition of PAO metabolism occurred in the low temperature case (10 °C). This study also experimentally demonstrated the loss of PAO metabolic ability in the subsequent aerobic stage, after the anaerobic stage. The presence of Cu(II) toxicity mainly resulted from the inhibition of biochemical reactions in the aerobic stage, and it was irrelevant to the inhibition of previous anaerobic metabolisms.

**Keywords:** polyhydroxyalkanoates (PHAs); copper ion; phosphorus removal; phosphorus accumulating organisms (PAO); temperature

## 1. Introduction

Enhanced biological phosphorus removal (EBPR) is considered an efficient process for removing phosphorus from wastewater. However, microorganisms are responsible for phosphate removal, and the mechanisms by which they accumulate phosphorus are complex and not well-understood [1]. This process remains operationally unstable in many systems, and it is difficult to reliably meet low effluent limits [2]. Substrate type has been identified as an important factor affecting the competition between polyphosphate accumulating organisms (PAOs) and glycogen accumulating organisms (GAOs), which are usually dominant in a deteriorated EBPR system [3].

Past research has also shown high inhibition of heavy metal ions to biological carbon, nitrogen, and phosphate removal in EBPR [4–8]. Reduction in EBPR efficiency was observed in a municipal wastewater treatment plant at total tin concentrations greater than 4 $\mu g L^{-1}$ in the solids fraction of the mixed liquor suspended solid (MLSS) [9]. Nevertheless, no influence on carbon or nitrogen removal efficiency was observed at this concentration. However, a possible mechanism for the Sn inhibition of PAOs was not provided. A related study was conducted by Tsai et al. [6]. They tried to chronically and stepwise add heavy metal Cd(II) ions into an $A_2O$ pilot plant to examine the toxic effects on organism behaviors. The results showed that an addition of 2 $mgCd L^{-1}$ started to affect the efficiency

of biological phosphate removal. About 76%, 64%, and 90% of the anaerobic P-release, the anoxic P-uptake, and aerobic P-uptake were inhibited, respectively. Wang et al. [10] investigated the shock load effect of Cu(II) on PAO metabolism at various pH and biomass levels, especially focusing on the variations of intracellular polyphosphate (poly-P), polyhydroxyalkanoate (PHA), and glycogen. Tsai et al. [11] further experimentally showed that PAOs losing the abilities of PHA production and phosphate uptake under Cu(II) presence came from the inhibition of the enzyme activities of acetyl-CoA synthases (ACS) and poly-phosphate kinase (PPK), respectively. This is because Cu(II) has a higher affinity for free ACS than for the ACS-coenzyme A complex. Another experiment also showed that free PPK is more readily bound to Cu(II) than the PPK-adenosine triphosphate complex.

The toxic effects of heavy metals on biological performances are complicated and correlated to other environmental conditions (e.g., pH, temperature, sludge retention time (SRT), carbon source, etc . . . ). You et al. [3] investigated the effect of Pb(II) on the dominance systems in PAOs and GAOs fed with acetic acid or glucose as the major carbon source, respectively. The results showed abnormal aerobic phosphate release was observed in the presence of Pb(II) in a high concentration of acetic acid. Tsai et al. [12] showed that the endurance of PAOs to Cu(II) highly related to SRT. PHA storage in the anaerobic stage for SRTs at 5 d and 15 d were more easily influenced by Cu(II) than that at 10 d SRT. The reaction of PAO phosphate uptake for the sludge at 10 d SRT retained half of the ability to take up phosphate in the presence of 2 mgCu L$^{-1}$, whereas PAO phosphate uptake was completely lost for SRTs at 5 d or 15 d. Temperature is considered one of the most important factors affecting the performance of phosphate removal in an EBPR process [13]. Generally, the wastewater temperature fluctuates according to the weather, and it is difficult to control well in practice. Unfortunately, the competition between PAOs and GAOs is highly related to temperature [14]. There were many studies regarding the effects of temperature on phosphate removal in the absence of heavy metals. However, the effects of temperature on P removal mechanisms and PAO metabolism are still unclear, and some are even contradictory as a result of incomparable experimental designs. Park et al. [15] observed a significant seasonal effect on EBPR in a wastewater treatment plant with poor EBPR efficiency during the summer, when the wastewater temperature was approximately 30 °C. At water temperature states less than 10 °C, EBPR was reported to be inhibited [16,17]. In some other studies [18,19], better EBPR performances were observed at lower temperature (5 °C) than at higher temperatures (10 and 15 °C). Similarly, EBPR efficiency has been reported to be improved at high temperatures (20–37 °C) in some studies [17,20], whereas better P removal efficiency was observed at lower temperatures (5–15 °C) in the other studies [21,22]. In the study performed by Panswad et al. [23], the PAOs were found to be lower-range mesophiles, or perhaps psychrophiles, and they predominated at 20 °C or possibly lower. The GAOs were somewhat mid-range mesophilic organisms with an optimum temperature between 25.0 °C and 32.5 °C. Whang and Park [24] observed PAOs were dominant at 20 °C in an anaerobic/aerobic (A/O) sequencing batch reactor (SBR) as a result of their higher anaerobic acetate uptake rates and aerobic biomass yields compared to GAOs. However, at 30 °C, GAOs were able to outcompete PAOs in the A/O SBR because of their higher anaerobic acetate uptake rates than PAOs. Ren et al. [25] reported that the influence of temperature shock on PAO abundance was more serious than that of GAOs in an enriching process. Following temperature shock from 22 ± 1 °C to 29 ± 1 °C and then to 14 ± 1 °C, GAOs were more temperature-adaptive and grew better than PAOs at 14 ± 1 °C.

There are some studies concerning the toxic effects of metal ions on phosphorus removal. However, how temperature variation influences the tolerance of PAOs and GAOs biomasses to metal ion toxicity is unknown. The long-term temperature effect is usually studied to examine the competition between PAOs and GAOs in activated sludge. On the contrary, studying the short-term temperature effect on a successful EBPR system can make the temperature effect more clear on the exact metabolisms of PAOs based on minimum interference of the GAO population. As we know, increasing temperature could speed up the metabolism of microorganisms. However, it is also possible to enhance the toxic effects of heavy metals on microorganism metabolisms. Temperature variations and toxic effects of

heavy metals are possibly encountered simultaneously in wastewater treatment plants and easily lead to the failure of biological process, resulting in an unstable status of EBPR system in practice. The metabolisms of biological phosphorus removal systems are quite a bit more complicated than biological carbon and nitrogen removal systems. The reasons for operational instabilities of EBPR, and how to control the system well, are still unknown. Especially, the paths in microorganism metabolisms that are easily affected by temperature and toxicity factors are not understood yet. Temperature and toxicity level variations might need to be considered together to completely understand their influences on biological phosphate removal mechanisms, especially the influence of the interaction between temperature and metal toxicity. Thus, the main purpose of this study was to study how temperature variation affects PAO tolerance to a toxic environment in the presence of copper ions, which is a very popular metal element used in real life and industries. The effluent standard of copper in wastewater is $3 \ mgL^{-1}$ in Taiwan, indicating that a copper level higher than $3 \ mgL^{-1}$ probably occurs in raw wastewater. And, the preliminary study (data not shown) showed low levels of copper ions could affect PAO behavior. Thus, this study chose $0$–$2 \ mgL^{-1}$ as the studied levels. Copper is a popular metal usually used in the lives of humans, and copper ions are easily detected in raw wastewater with noticeable concentrations. Hence, we chose copper as the target metal in this study. To exclude the interference of GAOs, temperature shock-loading experiments were conducted to study the effect of Cu(II) on PAO biomass metabolisms at different temperature levels (10, 20, and 30 °C), focusing on the anaerobic phosphate release, aerobic phosphate uptake, as well as the transformations of poly-P, intracellular PHAs, and glycogen in the activated sludge of a successful EBPR system. The results could be a practical reference for the optimal control of an EBPR process in a toxic environment with weather variations.

## 2. Materials and Methods

### 2.1. Sequencing Batch Reactor (SBR) Pilot Plant

An SBR pilot plant was used to acclimatize the activated sludge without copper ions. The total volume, base volume, and effective volume of the SBR were 160, 80, and 148.5 L, respectively. An operational cycle was 12 h, and there were 2 cycles per day. The phases and durations in a cycle were ordered as follows: anaerobic phase for 150 min, aerobic phase for 330 min, anoxic phase for 180 min, re-aeration phase for 10 min, sedimentation phase for 30 min, and decantation phase for 20 min. The SBR system was fed with a synthetic wastewater containing milk, $KH_2PO_4$, urea, $FeCl_3$, $CH_3COOH$, glucose, and $NH_4Cl$. The pH of the synthetic wastewater was adjusted to 6.8–7.2 with 6 N NaOH. The pH in the aerobic phase of the SBR system was automatically maintained at $7.2 \pm 0.2$ by adding an $NaOH/NaHCO_3$ solution, and the dissolved oxygen (DO) concentration in the aerobic phase was also controlled at $2.0 \pm 0.3 \ mgL^{-1}$. The SRT of the SBR plant was controlled at around 10 d. All batch experiments were conducted while the SBR reached a steady-state condition. Other details can be referred to You et al. [3].

### 2.2. Batch Experiments

A batch test was carried out in a 2 L acrylic vessel with a magnetic stirrer. To remove residuals of chemical oxygen demand (COD), phosphorus, and other chemical species, five (5) L of mixed liquor taken from the SBR pilot plant during the aerobic phase was washed twice using distilled water (DI) water. The washed solid was re-suspended in 5 L distilled water and was separated into six equal parts. $CuCl_2$ solution was then added into each batch reactor, and the pH was conditioned to 7.5. Acetate (HAc), which is the most abundant volatile fatty acid (VFA) species present in the influent of wastewater treatment plants, was added into the batch reactor as the only carbon source [26]. The concentration of acetate was $100 \ mg \ COD \ L^{-1}$. Nitrogen gas was aerated into the batch reactor to ensure an anaerobic status and then incubated in a water bath tank for 120 min with a constant

water temperature. Subsequently, the mixed liquid was aerated by air for 240 min to maintain an aerobic status.

*2.3. Analytical Methods*

The samples taken from the batch reactor were filtered by Millipore filter (0.45 µm pore size) before analyzing COD, total phosphorus (TP), $PO_4^{3-}$-P, MLSS, and mixed liquor volatile suspended solid (MLVSS) according to Standard Methods [27]. The PHA content in sludge was analyzed based on the method proposed by Chuang et al. [28]. The intracellular glycogen content in sludge was analyzed using the method described in Sudiana et al. [29]. An Atomic Absorption Spectrometry (Shimadzu AA-6200) was used to measure Cu(II) concentrations.

## 3. Results and Discussion

*3.1. Background-Temperature Effect on Enhanced Biological Phosphorus Removal (EBPR) without Cu(II)*

### 3.1.1. Anaerobic Metabolisms

Figure 1a shows the experimental results of the anaerobic metabolisms of PAOs at different temperatures. Both the specific phosphate release rate (SPRR) and specific substrate uptake rate (SSUR) of PAOs in the anaerobic stage increased with an increasing temperature from 10 to 30 °C. The results were similar to Panswad et al. [23], who studied the long-term temperature effect of EBPR at 20–35 °C. Nevertheless, the result of SSUR differed from Whang and Park [24], who found the SSUR in a PAO-enriched sludge increased as temperature increased from 10 to 20 °C, whereas the difference in the acetate uptake rate between 20 and 30 °C was insignificant.

In Figure 1a, the anaerobic PHA synthesis rates also increased with an increasing temperature, similar to phosphate release and acetate uptake, indicating the energy required for PHA synthesis of PAOs was mainly supplied by the hydrolysis of poly-P. Similar trends were found for the rates of glycogen degradation, which provided the reducing power for PHA synthesis, indicating PAOs provided more reducing power for PHA synthesis via degrading more glycogen while the temperature was increased.

Figure 1c shows the temperature effects on the ratios of P/HAc, PHA/HAc, and glycogen (Gly)/HAc in anaerobic conditions. The P/HAc values were nearly fixed around 0.204–0.226 P-mol/C-mol, regardless of temperature variation. Past studies obtained similar P/HAc values, e.g., 0.178–0.263 [23], 0.190 [30], and 0.132–0.173 [31]. This stoichiometric parameter is temperature-independent, as suggested by Brdjanovic et al. [31]. The Gly/HAc was also stable around 0.493–0.549 C-mol/C-mol, regardless of temperature. These Gly/HAc values were close to that proposed by Smolders et al. [30], who obtained a Gly/HAc value of 0.5 C-mol/C-mol in an enriched PAO culture system. The PHA/HAc values at 10, 20, and 30 °C were 1.18, 1.05, and 1.12 C-mol/C-mol, respectively. These PHA/HAc values were also close to that proposed by Smolders et al. [30], who obtained a PHA/HAc value of 1.33 C-mol/C-mol in an enriched PAO culture system. Brdjanovic et al. [31] obtained PHA/HAc value of 0.73–1.01 C-mol/C-mol in the temperature range 5–30 °C, and they also found a slight temperature effect on this ratio.

The PHA/Gly values, regarding the supply of reducing powers for PHA synthesis, at 10, 20, and 30 °C were 2.40, 1.96, and 2.04 C-mol/C-mol, respectively, as shown in Figure 1d. And, the PHA/P-released values, concerning energy supply for PHA synthesis, at 10, 20, and 30 °C were 5.77, 5.02, and 4.95 C-mol/P-mol, respectively.

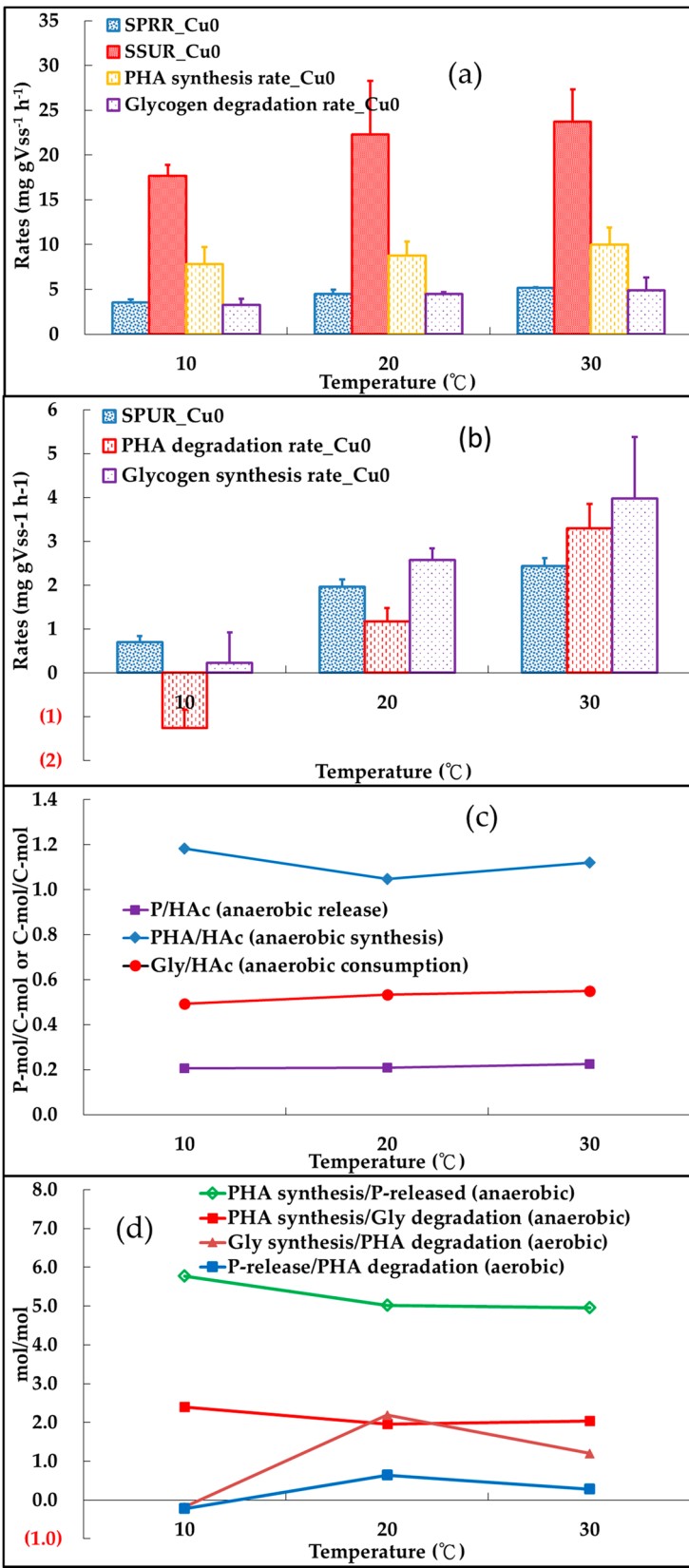

**Figure 1.** Relations between temperature and (**a**) the anaerobic metabolic rates; (**b**) the aerobic metabolic rate; (**c**) ratios of P-release/acetate (HAc), glycogen (Gly)/HAc, and polyhydroxyalkanoate (PHA)/HAc; (**d**) PHA synthesis/P-released, PHA synthesis/Gly degradation, Gly synthesis/PHA degradation, and P-release/PHA degradation.

### 3.1.2. Aerobic Metabolisms

Figure 1b shows the aerobic PAO metabolisms at different temperatures. Apparently, PAO metabolisms in the aerobic stage were much more influenced by temperature variations than at the anaerobic stage (by comparing Figure 1a,b). Both of the specific phosphorus uptake rate (SPUR) and PHA degradation rate were significantly increased with increasing temperature. These results were the same as the results reported by Brdjanovic et al. [31] in a studied range of 5–30 °C, but they were contrary to a long-term effect study reported by Panswad et al. [23]. Of note, the aerobic PHA degradation rate was negative at 10 °C, indicating PHA production rate was greater than PHA utilization rate in this condition. One possible reason was too much acetate remained at the end of the previous anaerobic stage because of low SSUR, resulting in PAOs that proceeded to synthesize PHAs under abundant carbon circumstances in the subsequent aerobic condition. Another possible reason was PAOs consumed less PHA because a 10 °C environment was unfavorable to the aerobic metabolisms of PAOs. The latter reason could be enhanced by lower SPUR and glycogen synthesis rates in a 10 °C environment compared to those in 20 and 30 °C conditions.

The glycogen synthesis rates in Figure 1b also increased with an increasing temperature, indicating the glycogen replenishment of PAOs was significantly promoted by increasing temperature in aerobic conditions. Interestingly, both rates of PAO phosphate uptake and glycogen synthesis were apparently low for the 10 °C trial, also indicating that a low temperature environment was unfavorable to the aerobic metabolisms of PAOs.

The amounts of phosphate uptake and glycogen synthesis per unit of PHA consumption are compared in Figure 1d. It could be used to evaluate the effects of temperature on PAO energy utilization, which were supplied by degrading PHAs, to take up phosphates and replenish glycogens in the aerobic stage. The results showed PAOs took up more phosphates and replenished more glycogens at 20 °C than in 10 and 30 °C environments, indicating 20 °C was beneficial for the aerobic metabolism of PAOs.

### 3.2. Temperature Influences on Anaerobic Metabolisms of Phosphorus Accumulating Organisms (PAOs) in Conjunction with Cu(II) Presence

### 3.2.1. Anaerobic Phosphate Release

Figure 2a,b shows that the presence of 0–2 $mgL^{-1}$ of Cu(II) had little impact on the metabolism of PAOs hydrolyzing poly-P for providing energy to take up substrate, i.e., SPRRs, regardless of temperature. Figure 2a also shows SPRRs increased with an increasing temperature, regardless of copper level, revealing no interaction between Cu(II) concentration and temperature on anaerobic phosphate release of PAOs.

### 3.2.2. Anaerobic Polyhydroxyalkanoate (PHA) Synthesis

Figure 2c shows that anaerobic PHA synthesis was markedly hindered with an increasing Cu(II) concentration, regardless of temperature. Figure 2d shows that the presence of Cu(II) at 30 °C had a higher inhibition percentage for PHA synthesis than in 10 and 20 °C trials, indicating the metabolism of anaerobic PHA synthesis of PAOs was more sensitive to Cu(II) presence at 30 °C than at 10 and 20 °C.

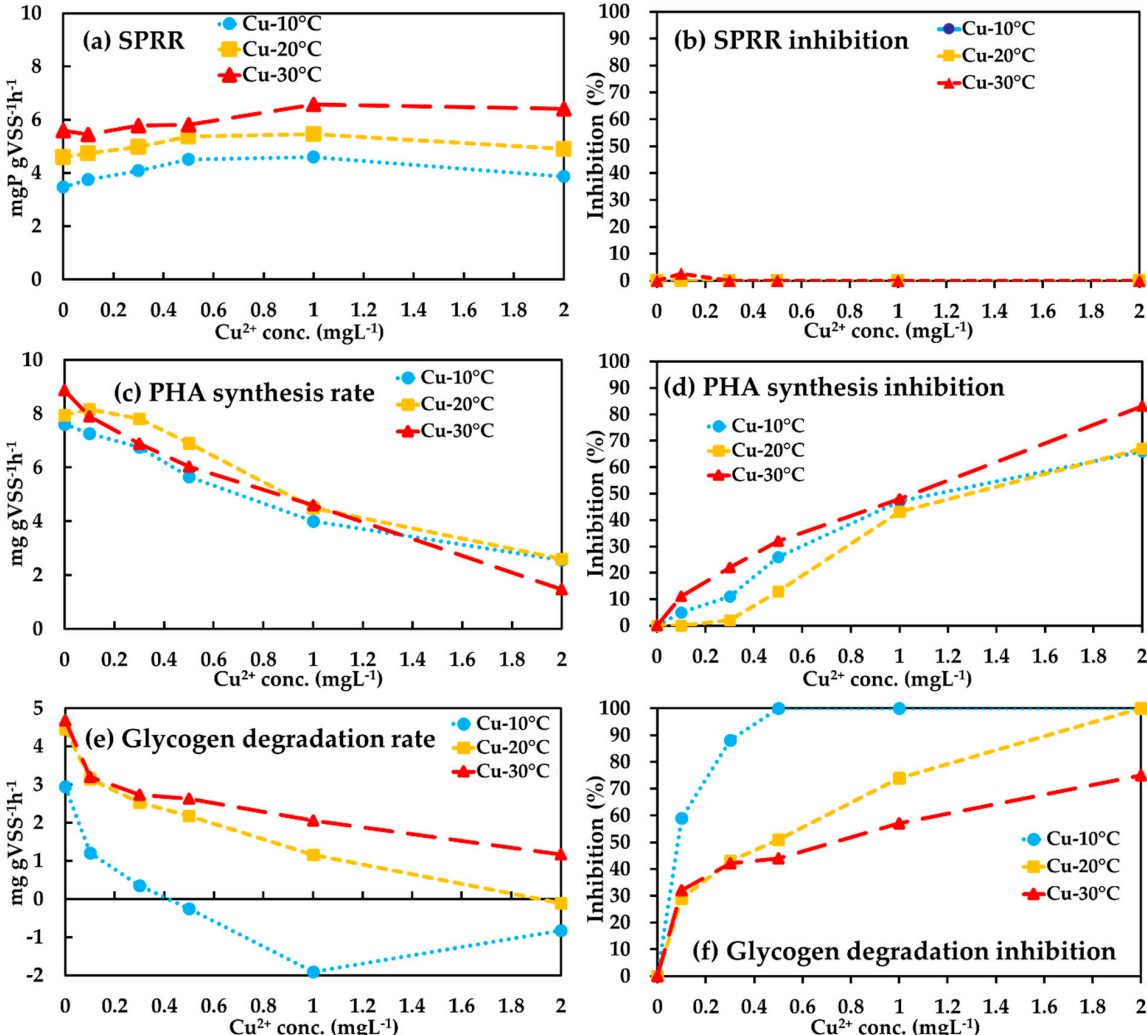

**Figure 2.** Toxic effects of Cu(II) on anaerobic specific phosphate release rate (SPRR), PHAs synthesis, and glycogen degradation under different temperature conditions. (**a**) SPRR; (**b**) SPRR inhibition; (**c**) PHA synthesis rate; (**d**) PHA synthesis inhibition; (**e**) Glycogen degradation rate; (**f**) Glycogen degradation inhibition.

### 3.2.3. Anaerobic Glycogen Degradation

As mentioned above, the presence of Cu(II) did not affect anaerobic phosphate release, but inhibited anaerobic PHA synthesis, regardless of temperature. The result indicated that the inhibition of anaerobic PHA synthesis of PAOs in the presence of Cu(II) was not a result of a lack of energy, which was normally provided by the hydrolysis of poly-P, necessary for synthesizing PHAs. There should be another important factor governing the biochemical reaction of anaerobic PHA synthesis when Cu(II) is present. Besides energy requirements, anaerobic PHA synthesis of PAOs demands a reducing power. About 70% of the reducing power relies on degrading glycogen to acetyl-CoA and oxidizing a partial number of them to $CO_2$ [32]. Obviously, glycogen degradation is essential for PHA synthesis and PAO proliferation, [33] and it is probably a limiting factor of anaerobic PHA synthesis of PAOs. Thus, we inferred that the most likely reason for decreased PHA synthesis in the presence of Cu(II) was due to the lack of reducing power provided by glycogen degradation. This inference could be further proved from Figure 2e,f, which show that glycogen degradation rates markedly reduced while increasing the Cu(II) concentration. Nevertheless, unlike the case of PHA synthesis in Figure 2c,d, the interaction between temperature and Cu(II) concentration on glycogen degradation was substantial. The lower the temperature, the larger the inhibition of glycogen

degradation by Cu(II) toxicity was. Especially in the 10 °C case, the presence of 0.5 mgL$^{-1}$ of Cu(II) could completely stop the process of anaerobic glycogen degradation of PAOs, as shown in Figure 2f. On the other hand, 2.0 mgL$^{-1}$ of Cu(II) completely inhibited the metabolism of glycogen degradation at 20 °C, whereas only 75% was inhibited at 30 °C. This result indicates PAOs in a high temperature environment (30 °C) had a good tolerance of Cu(II) with respect to anaerobic glycogen metabolism.

### 3.3. Temperature Influences on Aerobic Metabolisms of PAOs in Conjunction with Cu(II) Presence

#### 3.3.1. Aerobic Phosphate Uptake

Figure 3a shows the impacts of Cu(II) concentration on aerobic SPUR of PAOs at different temperatures. Unlike anaerobic phosphate release, the biochemical reaction of PAOs aerobically taking up phosphate with the addition of Cu(II) was remarkably inhibited. The temperature effect on such inhibition in conjunction with Cu(II) presence was apparent. The order of inhibition extent was 10 °C > 20 °C > 30 °C, as shown in Figure 3b. PAOs completely ceased phosphate uptake at 10 °C when 0.5 mgL$^{-1}$ of Cu(II) was present, whereas phosphate uptake ceased at 1.0 and 2.0 mgL$^{-1}$ at 20 and 30 °C, respectively.

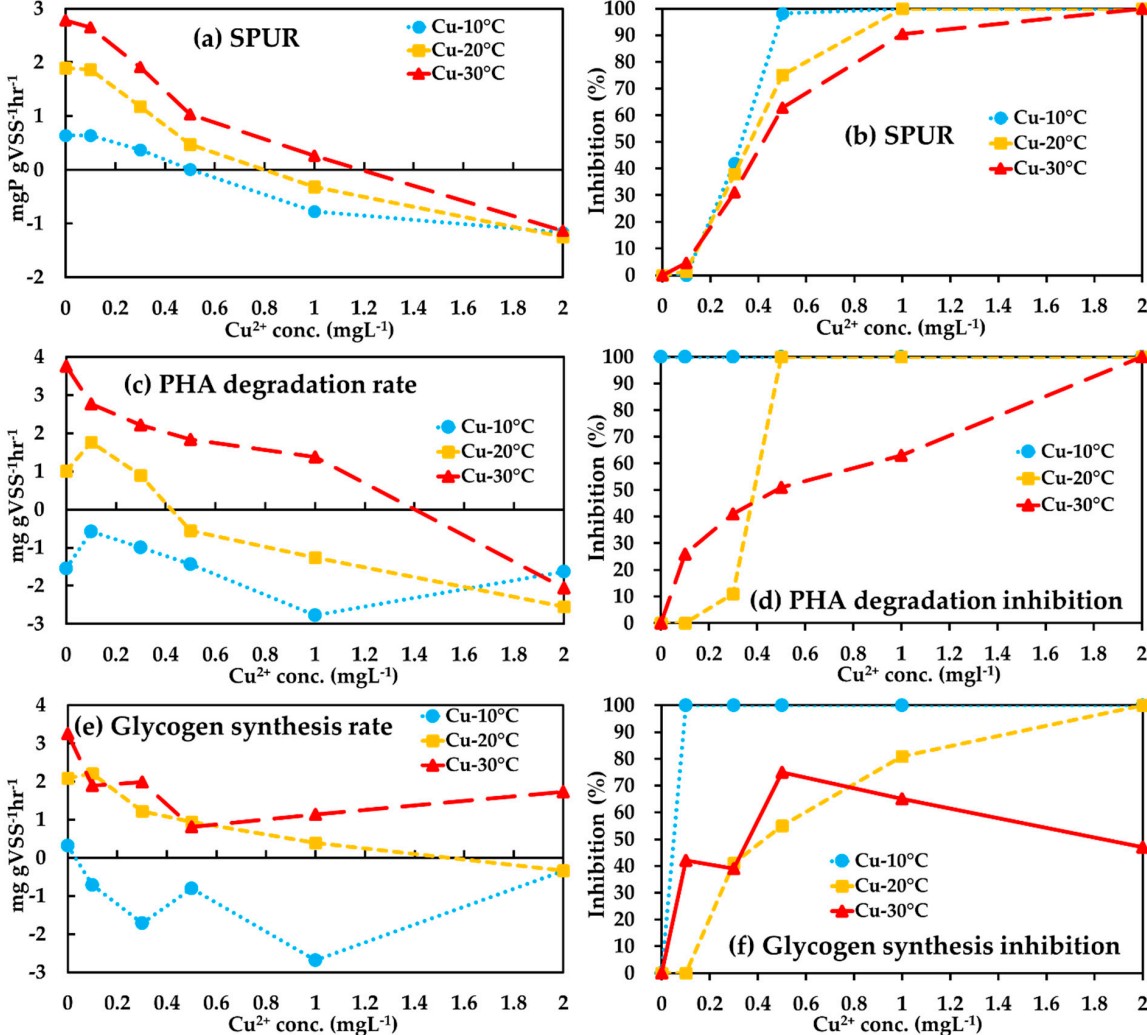

**Figure 3.** Copper toxic effects on aerobic specific phosphorus uptake rate (SPUR), PHAs degradation, and glycogen synthesis under different temperature conditions. (**a**) SPUR; (**b**) SPUR inhibition; (**c**) PHA degradation rate; (**d**) PHA degradation inhibition; (**e**) Glycogen synthesis rate; (**f**) Glycogen synthesis inhibition.

### 3.3.2. Aerobic PHA Degradation

Figure 3c shows the relations between the Cu(II) concentration and aerobic PHA degradation rate at different temperatures. Clearly, the aerobic PHA degradation rate reduced with an increasing Cu(II) concentration. The effect of temperature on the extent of metabolism inhibition in the presence of Cu(II) was also substantial. The order of inhibition extent was 10 °C > 20 °C > 30 °C, as shown in Figure 3d. PAOs completely stopped degrading PHAs at 20 °C when 0.5 mgL$^{-1}$ of Cu(II) was present, whereas PHA degradation stopped at 2.0 mgL$^{-1}$ at 30 °C, revealing PAOs had a higher tolerance of Cu(II) toxicity in a high temperature environment than in a low temperature environment. It is interesting that the PAOs synthesized PHA instead of consuming it in a 10 °C environment, regardless of Cu(II) levels, indicating a low temperature environment was unfavorable to the aerobic metabolism of PHA degradation.

### 3.3.3. Aerobic Glycogen Replenishment

Figure 3e shows the impacts of Cu(II) concentration on aerobic glycogen synthesis rates at different temperatures. Clearly, Cu(II) presence also influenced the aerobic glycogen replenishment of PAOs, regardless of the temperatures. Similar to the result of aerobic PHA degradation, temperature effect on the inhibition extent of glycogen replenishment was also obvious. The inhibition extent order was 10 °C > 20 °C > 30 °C, as shown in Figure 3f. PAOs completely ceased synthesizing glycogens at 20 °C when 2.0 mgL$^{-1}$ of Cu(II) was present, whereas it was almost 50% inhibition at 30 °C. Like aerobic PHA utilization, the PAOs at 10 °C consumed glycogen instead of synthesizing it, regardless of Cu(II) levels, indicating a low temperature was unfavorable to the aerobic glycogen metabolism.

### *3.4. Identification of Cu(II) Inhibition Extents in Anaerobic and Aerobic Stages*

To distinguish the degrees of Cu(II) inhibition of PAOs at different metabolic stages, two sets of experiments were conducted at 20 °C: (1) Cu(II) initially added into the reactor at the beginning of the first (anaerobic) stage, and (2) Cu(II) initially added into the reactor until the beginning of the second (aerobic) stage (i.e., at the end of the anaerobic stage).

### 3.4.1. Inhibition Causes of Phosphorus Metabolism

Figure 4a shows no significant difference between the cases of Cu(II) initially present in the anaerobic and aerobic stages, regardless of phosphorus release or uptake. This revealed that deterioration of the ability for PAOs to take up phosphorus at the aerobic stage directly originated from the inhibition of aerobic phosphorus metabolism, and it was irrelevant to the inhibition of the previous anaerobic phosphorus metabolism. The inhibition percentages of phosphorus metabolisms were differentiated from different metabolic stages, as shown in Figure 4b. At 0.5 mgL$^{-1}$ of Cu(II), no inhibition of the anaerobic phosphorus release was found, whereas 36% of the aerobic phosphorus uptake was inhibited, among which, only 5% of inhibition came from the previous anaerobic stage and 31% resulted from the inhibition at the aerobic stage itself. At 1.0 mgL$^{-1}$ of Cu(II) only 5% of the anaerobic phosphorus release was inhibited, whereas 82% of the aerobic phosphorus uptake was inhibited because of the inhibition at the aerobic stage itself. At 2.0 mgL$^{-1}$ of Cu(II), 100% of the aerobic phosphorus uptake was inhibited because of the inhibition at the aerobic stage itself.

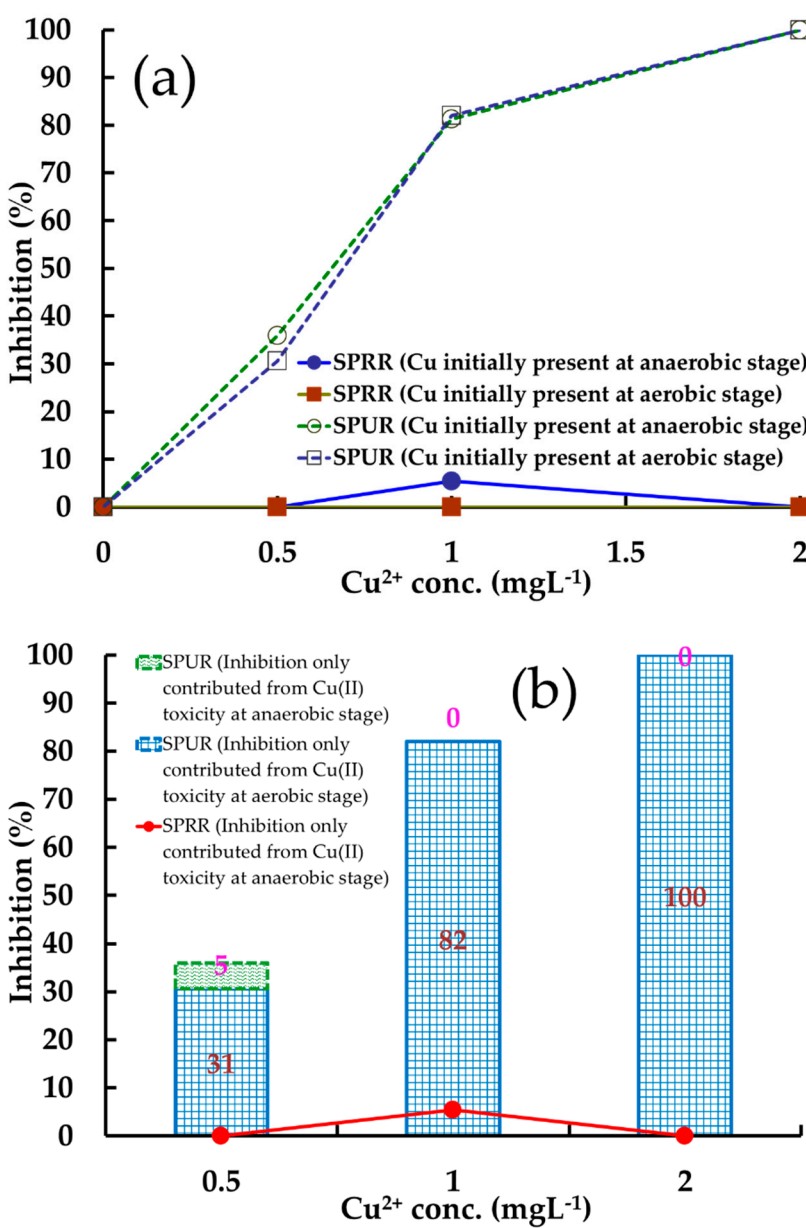

**Figure 4.** Cu(II) inhibition of phosphorus metabolisms at 20 °C. (**a**) Cu(II) appearance at different stages; (**b**) inhibition contributions from different stages.

### 3.4.2. Inhibition Causes in PHA Metabolism

Unlike phosphorus metabolism, Figure 5a shows that 38–56% of the anaerobic PHA synthesis was inhibited when 0.5–2 mgL$^{-1}$ of Cu(II) was added at the anaerobic stage. Meanwhile, no significant difference was found for the inhibitions of aerobic PHA degradation between the cases of Cu(II) initially present in anaerobic and aerobic conditions. This result indicated a deterioration in the ability of PAOs to utilize PHAs in aerobic conditions, resulting from the inhibition of aerobic PHA metabolism itself, and it was irrelevant to the inhibition of the previous anaerobic PHA metabolism. The inhibition percentages of PHA metabolisms were differentiated from different metabolic stages, as shown in Figure 5b. At 0.5 mgL$^{-1}$ of Cu(II), 38% of the anaerobic PHA synthesis was inhibited, whereas 48% of the aerobic PHA degradation was inhibited from the Cu(II) toxicity at the aerobic stage itself. At 1.0 mgL$^{-1}$ of Cu(II), 48% of the anaerobic PHA synthesis was inhibited, whereas 80% of the aerobic PHA degradation was inhibited, among which, only 2% of inhibition came from the previous anaerobic stage, and 78% was ascribed to the inhibition at the aerobic stage itself. At 2.0 mgL$^{-1}$ of Cu(II), 56%

of the anaerobic PHA synthesis was inhibited, whereas 80% of the aerobic PHA degradation was inhibited from the Cu(II) toxicity at the aerobic stage itself.

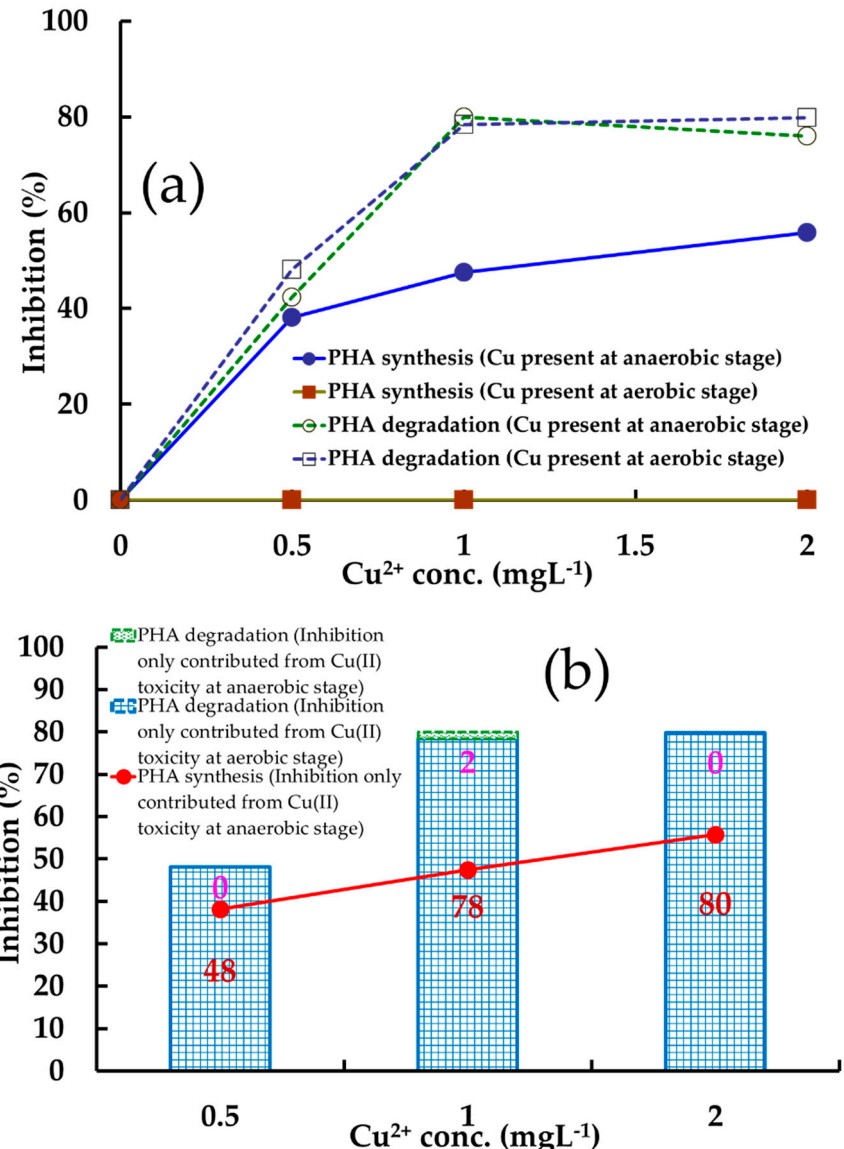

**Figure 5.** Cu(II) inhibition of PHA metabolisms at 20 °C. (**a**) Cu(II) appearance at different stages; (**b**) inhibition contributions from different stages.

## 4. Conclusions

This study showed temperature variation indeed affected the inhibitive degrees for PAO metabolism in the presence of Cu(II) toxicity. The deteriorations of most PAO metabolisms were amplified in a low temperature (10 °C) environment. PAOs had a higher tolerance of Cu(II) toxicity in a high temperature environment. This study further proved that the loss of the ability of PAOs to take up phosphorus and utilize PHAs in the aerobic stage directly originated from the inhibition of aerobic metabolism, and it was irrelevant to the inhibition of previous anaerobic metabolisms. This study experimentally identified the interaction between temperature and copper metal toxicity on complex mechanisms of biological phosphate removal. It is especially novel that the study distinguished the degrees of Cu(II) inhibition of PAOs at different metabolic stages. The results help us to more deeply understand the reasons why biological phosphate removal processes remain operationally unstable in

many systems. Thus, it is difficult to reliably meet low effluent limits. Based on the results of this study, it is easier to control EBPR by considering the variations of temperature and metal toxicity in practice.

**Author Contributions:** Conceptualization, Y.-P.T. and C.-C.Y.; methodology, C.-C.C. and M.-S.L.; formal analysis, L.-Y.H.; resources, Y.-P.T.; writing—original draft preparation, Y.-P.T.; writing—review and editing, C.-C.Y., and C.-C.C.; supervision, C.-C.C. and C.-C.Y.; project administration, M.-S.L.; funding acquisition, Y.-P.T. and C.-C.C.

**Funding:** This study was supported by a grant from the Ministry of Science Technology, Taiwan, Republic of China (MOST 106-2221-E-260-003-MY3).

**Conflicts of Interest:** The authors declare no conflict of interest.

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
