# Peer review of "The Influence of Temperature on Metabolisms of Phosphorus Accumulating Organisms in Biological Wastewater Treatment Plants in the Presence of Cu(II) Toxicity"

_applsci, doi:10.3390/app9061126_

Round 1
Reviewer 1 Report
The paper titled “The influence of temperature on metabolisms of phosphorus accumulating organisms in the presence of Cu (II) toxicity” deals with an experimental study aimed at assessing the effect of temperature on the metabolism of phosphorus accumulating organisms (PAOs) in an environment made toxic by presence of Cu.
The English language is properly used and the paper is well written.
Nevertheless, the paper is not suitable for publishing in its current version and I invite the authors to highlight the novelty of the work and clarify its usefulness, because it is known that increasing the temperature the metabolism of microorganisms goes faster as long as it is reached a specific value of temperature when microorganisms start to die. And the second point that needs to be clarified is why the authors have considered an environment with high concentration of copper. Have they considered a real and common wastewater? Why have they chosen Cu and not other potential toxic elements (PTEs)? Or a combination of them?
Specific comments
-line 167. Leave a space after 2;
-line 220. Delete a point (.) after “concentration”;
-line 259. Check the style for subsection heading;
-lines 301-306. Extend the conclusion section pointing out the novelty and usefulness of the work.
Check the size for all figures.
Author Response
Point 1:Nevertheless, the paper is not suitable for publishing in its current version and I invite the authors to highlight the novelty of the work and clarify its usefulness, because it is known that increasing the temperature the metabolism of microorganisms goes faster as long as it is reached a specific value of temperature when microorganisms start to die. And the second point that needs to be clarified is why the authors have considered an environment with high concentration of copper. Have they considered a real and common wastewater? Why have they chosen Cu and not other potential toxic elements (PTEs)? Or a combination of them?
Response 1:Thanks a lot for reviewer’s comment. The manuscript has been revised. (1) We highlight the novelty of the work and clarify its usefulness at Lines 89-100. (2) We explain the reasons why we chose high concentration of copper as target metal at Lines 102-108.
Point 2:Specific comments
Response 2:Thanks a lot for reviewer’s comment. The manuscript has been revised. (1) We have already leave a space after 2 at Line 183.
(2) We have already delete a point (.) after “concentration” at Line 236.
(3) We have already modified the style for subsection heading at Line 275.
(4) We have extended the conclusion section pointing out the novelty and usefulness of the work at Lines 322-329.
(5) We have already checked the size for all figures. Figures 4 and 5 become smaller.

Reviewer 2 Report
The manuscript “The influence of temperature on metabolisms of phosphorus accumulating organisms in the presence of Cu(II) toxicity” describes and analyzes the toxicity effects of Cu(II) at the variation of temperature at 10° C, 20° C and 30° C. The paper is scientifically sound and valid, and will increase the knowledge in the field of enhanced biological phosphorus removal. Nevertheless, some modifications and improvements to increase the quality of the paper are necessary before it can be accepted for publication in Applied Sciences. Please refer to my comments below.
1) L36-37: Please rephrase the sentence.
2) L102: please correct the sentence “there was 2 cycles a day” with “there were 2 cycles per day”
3) L104-105: How it is possible to have a continuous inflow in an SBR? Shouldn’t the dosage of the influent stop after the initial filling step? In such a scenario, the reactor cannot be identified as CSTR but batch, as the name SBR suggests.
4) Please use the correct notation with chemical substances. E.g., L107 CH3COOH L109 NaHCO3
5) Figure 1, 2, 3, 4b, 5b: I would suggest to use colors in the figures; the paper’s quality would greatly benefit
6) Please doublecheck the manuscirpt for typos.
Author Response
Point 1:L36-37: Please rephrase the sentence.
Response 1:Thanks a lot for reviewer’s comment. The sentence at L35-37 has been rephrased as “Reduction in EBPR efficiency was observed in a municipal wastewater treatment plant at total tin concentrations greater than 4 μgL−1 in the solids fraction of the mixed liquor suspended solids [9].”
Point 2:L102: please correct the sentence “there was 2 cycles a day” with “there were 2 cycles per day”
Response 2:Thanks a lot for reviewer’s comment. The manuscript has been revised. Please see Line 118.
Point 3:L104-105: How it is possible to have a continuous inflow in an SBR? Shouldn’t the dosage of the influent stop after the initial filling step? In such a scenario, the reactor cannot be identified as CSTR but batch, as the name SBR suggests.
Response 3:Thanks a lot for reviewer’s comment. The manuscript has been revised. We have remove the sentences “The inflow pattern was continuous and the system could be modeled as a continuously stirred reactor”. Please see Lines 121-122.
Point 4:Please use the correct notation with chemical substances. E.g., L107 CH3COOH L109 NaHCO3.
Response 4:Thanks a lot for reviewer’s comment. The manuscript has been revised. Please see Lines 123 & 125.
Point 5:Figure 1, 2, 3, 4b, 5b: I would suggest to use colors in the figures; the paper’s quality would greatly benefit
Response 5:Thanks a lot for reviewer’s comment. The manuscript has been revised. All figures are colorful.
Point 6:please doublecheck the manuscirpt for typos.
Response 6:Thanks a lot for reviewer’s comment. The manuscript has been carefully checked again.
phosphorus accumulating organisms in biological wastewater treatment plants in the presence of Cu(II) toxicity】. We also have add some broader implications in the conclusions. Please see Lines 322-329.

Reviewer 3 Report
The manuscript deals with the influence of temperature on metabolisms of phosphorus accumulating organisms in the presence of Cu(II) toxicity. The current version of your manuscript is not a scientific communication. Article is really poor prepared, all figures are vague. Link to novelty is weak. Lines 298-299: Figure 5. Cu(II) inhibition of PHA metabolisms at 20°C (a) Cu(II) appear at different stages, the font of this figure should be the same as the rest of the text, the same situation is with the rest of the figures, it should be improved. Lack of affiliations of all Authors. Affiliation 1; ccc6166@yahoo.com.tw 6 2 Affiliation 2; s96322902@mail1.ncnu.edu.tw 7 3 Affiliation 3; mslu@ncnu.edu.tw 8 4 Affiliation 4; whattodo38@gmail.com 9 5 Affiliation 5; yptsai@ncnu.edu.twm
The hypothesis or purpose of the work should be rewritten in more clearly way. Also the introduction should include information about problem, as well as reasons for conducting the research. The manuscript does not include the value added with respect to existing research. Authors do not explain well, where is the novelty of the distinguished research. Compare the obtained achievements with previous studies in details. Conclusion is not sufficiently described. It is more like summary of information, what can be read in previous chapters. How in practice the results of the presented work can be used? This should be discussed in the point concerning discussion of the results. Besides, there is no discussion about possible limitations of using the proposed modelling. The last point of the article contains in fact only the conclusions relating to the performed analysis, but there is no more detailed perspective. Some information about practical use of the obtained results, both in the section Results and Conclusions should be underlined. The impression occurs, that the conclusions were formulated very sharply beforehand, what weakens the conclusions.
Author Response
Point 1:Article is really poor prepared, all figures are vague. Link to novelty is weak. Lines 298-299: Figure 5. Cu(II) inhibition of PHA metabolisms at 20°C (a) Cu(II) appear at different stages, the font of this figure should be the same as the rest of the text, the same situation is with the rest of the figures, it should be improved. Lack of affiliations of all Authors. Affiliation 1; ccc6166@yahoo.com.tw 6 2 Affiliation 2; s96322902@mail1.ncnu.edu.tw 7 3 Affiliation 3; mslu@ncnu.edu.tw 8 4 Affiliation 4; whattodo38@gmail.com 9 5 Affiliation 5; yptsai@ncnu.edu.twm
Response 1:Thanks a lot for reviewer’s comment. The manuscript has been revised.
(1) All figures are re-plotted and presented by colors. They become more clear than before.
(2) The font of all figures are the same as the text.
(3) The affiliations have been added.
Point 2:The hypothesis or purpose of the work should be rewritten in more clearly way. Also the introduction should include information about problem, as well as reasons for conducting the research. The manuscript does not include the value added with respect to existing research. Authors do not explain well, where is the novelty of the distinguished research. Compare the obtained achievements with previous studies in details. Conclusion is not sufficiently described. It is more like summary of information, what can be read in previous chapters. How in practice the results of the presented work can be used? This should be discussed in the point concerning discussion of the results. Besides, there is no discussion about possible limitations of using the proposed modelling. The last point of the article contains in fact only the conclusions relating to the performed analysis, but there is no more detailed perspective. Some information about practical use of the obtained results, both in the section Results and Conclusions should be underlined. The impression occurs, that the conclusions were formulated very sharply beforehand, what weakens the conclusions.
Response 2:Thanks a lot for reviewer’s comment. The manuscript has been revised according to this review’s suggestion. The purpose of the work has been more clear. The novelty is also enhanced in the revised manuscript. Conclusion section is also enhanced. Please see Lines 88-107 and Lines 321-328.
Academic Editor Comments:
Point 3:The title needs to be rephrased to make the connection between wastewater and storm water treatment more apparent. Also, please include some additional information in the conclusions to discuss broader implications of your experimental results.
Response :Thanks a lot for reviewer’s comment. The manuscript has been revised. The title has been changed to 【The influence of temperature on metabolisms of phosphorus accumulating organisms in biological wastewater treatment plants in the presence of Cu(II) toxicity】. We also have add some broader implications in the conclusions. Please see Lines 322-329.

Round 2
Reviewer 3 Report
My numerous remarks were included in the manuscript, therefore I accept it.